# Pressure-Driven Micro-Casting for Electrode Fabrication and Its Applications in Wear Grain Detections

**DOI:** 10.3390/ma12223710

**Published:** 2019-11-10

**Authors:** E Cheng, Ben Xing, Shanshan Li, Chengzhuang Yu, Junwei Li, Chunyang Wei, Cheng Cheng

**Affiliations:** 1School of Mechanical Engineering, Hebei University of Technology, Tianjin 300132, China; 2State Key Laboratory of Reliability and Intelligence of Electrical Equipment, Hebei University of Technology, Tianjin 300130, China; 3Hebei Key Laboratory of Robotic Sensing and Human-robot Interactions, Hebei University of Technology, Tianjin 300132, China; 4Department of Computer Science and Electrical Engineering, Hebei University of Technology, Langfang 065000, China; 5Institute of Biophysics, School of Science, Hebei University of Technology, Tianjin 300401, China; 6School of Engineering and Computer Science, Morehead State University, Morehead, KY 40351, USA

**Keywords:** microfabrication, melting metal, surface pretreatment, impedance cytometer, differential amplifier, wear particle detection

## Abstract

The microelectrode is an essential and vital part in microsensors that are largely used in industrial, chemical, and biological applications. To obtain desired microelectrodes in great quality, it is also of great necessity and significance to develop a robust method to fabricate the microelectrode pattern. This work developed a four-terminal differential microelectrode that aims at recognizing microparticles in fluids. This microelectrode pair consisted of a high height–width ratio microelectrode array fabricated using a pre-designed microelectrode pattern (a micro-scale channel) and melted liquid metal. The surface treatment of microelectrodes was also investigated to reveal its impacts on the continuality of melting metal and the quality of the fabricated microelectrode patterns. To evaluate the performance of micro-casting fabricated electrodes, a microfluidic device was packaged using a microelectrode layer and a flow layer. Then impedance cytometer experiments were performed using sample fluids with polymer particles in two different sizes in diameter (5 μm and 10 μm). In addition, engine oil was tested on the microelectrodes as complex samples. The number of abrasive particles in the engine oil can be collected from the developed microfluidic device for further analysis.

## 1. Introduction

Microfabrication techniques offer a great potential for micro-sensing applications [1,2] due to the fact of their compact size and high sensitivity. In this context, a major challenge for microelectrode fabrication is to develop cost and time efficient methods for the fabrication of a robust conductive layout. The most commonly used techniques for microfabrication are the top-down and bottom-up [3] methods. The top-down fabrication technique, such as wet/dry-etching [4], is based on patterning on large scale while reducing the lateral dimensions to the microscale. The bottom-up fabrication technique, such as 3D printing [5], arranges atoms and molecules in microstructures. The bottom-up fabrication is a promising approach which directly deposits conductive metal or polymers onto a substrate. Recently, the field of stretchable/flexible electronics has been exploring the use of soft materials to develop microelectrode arrays [6,7]. It overcomes the inflexibility of conventional microelectronics devices that are typically fabricated on silicon wafers. Metal printing is one of the addictive microfabrication approaches that allows for the single-step fabrication of conducting circuits [8]. Directly printed flexible electrode arrays shows a great potential for wearable device applications. However, the conductive inks often require specific synthetics procedures [9]. Inspired by the casting [10] technique which is well used for macro-scale fabrications, the micro-casting of an electrode array was investigated herein. One key issue of microelectrode casting is to load conductive metal liquid into the channel molds. Generally, it is not easy to load liquid samples into microchannels which often have a big flow resistance or high surface energy. For instance, a syringe pump or external air source is often required to pump liquids into the microchannels for most microfluidic applications. In order to load water-based reagents into a microchannel smoothly, a plasma hydrophilic treatment is suggested to change the surface energy of fluid–channel interface. However, with regard to studies on the sample loading [11,12] in microchannels, currently there are no works on the surface treatment of melting metal flows in a microfluidic system. The surface energy of a metal–channel interface is quite different from those of a water–channel interface; thus, experimental studies should be done to improve the metal liquid loading performance. This work focuses on the effects of surface treatment and inlet geometrical parameters on micro-casting fabrication performance. 

The fabrication procedures, as well as the capability to perform electric signal-based detections are the main concerns of this work. The two main factors that can characterize the micro-casting and electrode formation are: (1) the surface energy of the inner walls and (2) the size of the inlet which is the main entrance to load liquid metal into the microchannels. A detailed study was conducted to investigate the role of these factors on fabrication performance. There was no external power utilized to pump the fluidic metal into the microchannels because the metal was in solid status until it was heated at the melting point. Pressure-driven (positive pressure or negative pressure) flow is used to load the melting metal into microchannels. As an indirect soft tooling process, vacuum casting is a copying technique characterized by the use of negative pressures during the fabrication of patterns [13]. However, in order to prevent blocking of the nozzles of the vacuum machine, it is not recommended to carry out micro-casting using a vacuum. Then, the positive pressure provided by the atmospheric pressure could provide energy for microfluidic pumping within the channel. In other words, the atmospheric pressure-driven micro-casting for electrode fabrication was presented, and its performance was validated by the classification and counting of standard latex particles. Moreover, abrasive grains in the engine oil were also detected, showing good performance of the micro-casting-fabricated electrode arrays.

The master mold pattern used in this work was a Christmas-tree like channel which is often used as a linear gradient concentration maker [14]. The complex channel network design consisted of a tree-like architecture at each channel branch. Although the Christmas-tree pattern is not popular as an electrode layout, we still preferred using it to design and fabricate the complex channel network because it is more difficult to fabricate. In Christmas-tree network devices, the conductive material experiences heating and melting, atmospheric pressure-driven pumping, flow through all the branches to reach the outlet, cooling, and finally formation of the ending terminal of the electrode. Next, its application in particle classification/counting was carried out. Standard latex particles with different diameters were distinguished from each other. Also, the metallic wear particles in engine oil were calculated using four-terminal electrodes connected with an impedance analyzer. Therefore, the fabrication technique notes presented in this paper can maximize the probability of rapid fabrication of high height–width ratio microelectrodes. In addition, the information provided in this work can guide researchers to fabricate electrode devices with their desired dimensions for MEMS devices.

## 2. Material and Methods

### 2.1. Electrode Fabrication by Pressure-Driven Micro-Casting Techniques

Figure 1 demonstrates the procedures of the micro-casting fabrication. Using the soft photolithography [15] process, the pattern was cured out of SU8 photoresist on a silicon wafer or glass substrate. With a width of 100 µm and height of 30 µm, the SU8 master and PDMS (polydimethylsiloxane) channel were soft lithography fabricated, as shown in Figure 1a. After peeling off from the previous substrate, the patterned PDMS layer was in-reversable bonded with a glass slider. Then, the inner wall of the PDMS-glass channel surface was treated, as demonstrated in Figure 1b. The reagent for surface treatment was fresh prepared by γ Pyrenees trimethoxysilane (98%; CAS:4420-74-0; Aladdin, Shanghai, China) and Acetylene (99.5%; CAS75-05-8; Aladdin, Shanghai, China) in 1:10 v/v%. The purpose of this step was to obtain proper surface energy for adhesive bonding between the channel and melting metal. The surfactant was injected into the channel and kept for 3–5 minutes and followed by an air flush. Next, the treated microfluidic device was put onto a hot plate at 180 °C for 5 minutes. A Sn42Bi58 alloy solder wire (0.6 mm in diameter; Sn, 42% and Bi, 58%; melting point: 138 °C; Shenzhen Baoda Tin Material Inc., Shenzhen, China) was carefully inserted into the inlet of the treated microfluidic device. Then, the solder wire was melted into metal liquid in seconds, as shown in Figure 1c. Once the PDMS-glass channel was filled with metal liquid, the solder wire was cut at the outlet of the channel. The device was then removed from the hotplate to cool at room temperature. The electrical connection check was conducted using a multimeter to make sure the solder had a good connection. Then, the electrode pattern was peeled off the slider and bonded onto a new glass slider using a plasma bonding machine (PTL-VR500, PTL Electrical Technology, Zhaoyuan, China). As shown in Figure 1d, the microelectrode pattern was transferred onto the glass substrate. The micro electrode width and height depended on the PDMS channel which functions as a casting mold. Accordingly, the electrode width was 100 µm, and the height was 30 µm. Thus, the height–width ratio was 30% in this work. Compared with the height–width ratio (~0.1%) of the wetting-etched microelectrodes (~100 µm in width, ~100 nm in height), the height–width ratio of the microelectrodes fabricated by the micro-casting technique was very high.

### 2.2. Procedures for Particle Classification/Counting Applications

Two kinds of latex particles (10 μm and 5 μm, Sigma-Aldrich, St. Louis, MO, USA) were prepared in deionized water. The concentration of 10 μm particle was 1.2 × 10^5^ count/mL while the concentration of 5 μm particles was 1.8 × 10^5^ count/mL. The concentration was calculated by microscopic counting method. A mixture of 10 μm and 5 μm particles (1:1 v/v%) were prepared as the reagent for proof of concept experiments. The mixture was pumped into the microfluidic device at the flow rate of 20 μL/min. The four-terminal electrode array was electrically connected with an impedance analyzer (HF2IA, Zurich Instruments, Zurich, Switzerland) and a differential amplifier. Then, the real-time current was recorded by the impedance analyzer.

## 3. Results and Discussions

### 3.1. Effect of the Suface Treatment on the Micro-Casting Performance

Surface modifications are effective treatments for enabling the bonding of polymeric microfluidic devices [16]. For the pumping of water into PDMS microfluidic devices, plasma treatment [17,18] of glass/silicone surfaces is an environmentally sound method to increase wettability and improve adhesion abilities. However, the plasma treatment did not work well to improve the pumping of liquid metals. The reason is that the main purpose of plasma treatment is to make the PDMS inner walls more hydrophilic. Then water-based liquid can be loaded into the PDMS microchannels easily. However, the plasma had no effect on the metallophilic treatment. In order to load melting metal liquid into the PDMS microchannel, the chemical treatment was applied. The effect of surface treatment on the micro-casting fabrication was tested for the first time as shown in Figure 2. Figure 2a shows a typical image of a PDMS-glass casting mold obtained by reversable bonding of a glass slider and a PDMS microchannel fabricated by soft lithography techniques. Two representative failures are also demonstrated in Figure 2b, c. If the inner wall of the microchannel mold is not treated before loading the solder wire, it is not easy to fill the channel completely, as shown in Figure 2b. Although the metal liquid could reach the outlet of the microchannel, here were still had many “blind regions” where the metal liquid could not be trapped within the channel. Besides, as shown in Figure 2c, metal residues were observed after peeling off the metal-in-PDMS layer from the glass substrate. Also, the present reagent mixture ratio, described in sub-Section 2.1, was an important factor to peel off the electrodes successfully. If these agents were not mixed in 1:10 v/v%, the failed “blind regions”, shown in Figure 2c,d, could also be observed. On the other hand, the advantages of surface treatment were also validated with a good fabrication performance. If the inner walls were well treated by the mixed reagents, then the PDMS microchannel mold could be filled with melting metal as expected (Figure 2d). A typical microscopic image of the microelectrode profiles shows that the geometry and dimension (~100 μm) of the electrodes fabricated by micro-casting met the requirements of most applications. Also, the metal-in-PDMS layer was easily peeled off from the glass substrate without any residue as shown in Figure 2e.

### 3.2. Effect of the Inlet Dimensions on the Micro-Casting Performance

The effect of inlet dimensions on the micro-casting fabrication performance was also investigated as shown in Figure 3. Figure 3a shows an image of a PDMS-glass casting mold. It was obtained by reversable bonding of a glass slider and a patterned PDMS microchannel fabricated by soft lithography techniques. The Christmas-tree like pattern consisted of one inlet and five independent outlets. It is important to note, here, that the inlet is usually utilized as the outlet in linear concentration microfluidic applications. As demonstrated in Figure 3b, the pattern of the inlet was designed as large as 2 mm in diameter. However, the size of the punched inlet hole depends on the connecting tube size (from hundreds of microns to several millimeters), thus it is practical to make inlet holes of different sizes. In this work, inlet holes of 0.6 mm and 1.0 mm were fabricated by a blunt puncher. Figure 3c, d show the micro-casting fabrication performance using 1 mm and 0.6 mm inlet holes, respectively. This section covers some tips for metal-in-PDMS microelectrode fabrications. Bubbles are crucial problems for flow stability and continuality. Effective operation of bubble prevention during micro-casting fabrication will help us achieve better fabrication qualities. The bubble risk could be best avoided by decreasing the inlet dimensions. Using this method, no bubbles or dis-continuality was observed in the micro-casting of melting solder wires. The examples in Figure 3c, d showed the manners in which, by decreasing the inlet size, the flow control performance of melting metal within the PDMS channel can be improved. Thus, it is possible to obtain a well patterned electrode by micro-casting technique. 

### 3.3. Proof of Concept: Application in Latex Particle Counting 

To ensure that the microelectrode was well fabricated by the micro-casting technique, a four-terminal interdigitate electrode array was designed and casted as shown in Figure 4. As an application example, the proof-of-concept application was taken for particle counting. The classification and counting of particles (i.e., cells, abrasive grains, etc.) can be achieved in many ways [19,20]. Reference [19] provides a review of the principle of particle analysis from impedance information. Optical detection particle counters based on the optical density [21], microscopic detection by image processing [22], or electrical detection based on the real time electrical signals [23] (i.e., voltage, current, or impedance) have been developed and studied. As shown in Figure 4, the device was packaged from two independent PDMS layers. The top layer, which had a metal-in-PDMS architecture, had four independent electrode terminals. The top layer was a flexible layer in which a four-terminal metal electrode pattern was embedded, while the bottom layer was designed for particle flows passing through the four-terminal electrodes. The particle flows within the bottom PDMS layer had three subchannels: the central one was used for particle flow sample loading, while the other two branches were for sheath flows. They were plasma bonded to make a microfluidic device. The electrodes on the top layer were electrically connected to the AC signal and a differential amplifier from the impedance analyzer. In order to connect tubing from the top layer, all the outlets and inlets of the bottom fluidic layer were designed as through-holes. In order to fabricate the four-terminal interdigitated microelectrode in the top PDMS layer, eight ports (four inlets and four outlets; each “inlet–outlet” pair is demonstrated as i, ii, iii, and iv in Figure 4) were pre-designed for micro-castings as demonstrated by the yellow, dashed lines in Figure 4. The bottom layer consists of a “cross”-like sheath flow channel (the blue, dashed lines in Figure 4), three inlets, and one outlet. 

The detection of abrasive grains in engine oil was challenging, because the size, material, and shape of these grains were unknown before the experimental tests were taken. Here, we took an observation of the particle classification and counting as a proof of concept, to show the performance of microelectrodes fabricated by pressure-driven micro-casting techniques. 

As a first step, a set of experimental tests was conducted to compare the experimental particle counts with the estimated particle numbers. Figure 5a shows the real-time current signal obtained by the impedance analyzer. There would be a current pulse I_s_(t) when the particle is passing through the electrodes. The assumption here is shown in Equation (1), as the following,
I_n_(t) < I_s_(t)/3(1)
where I_n_(t) is the background noise and I_s_(t) is the signal generated by the particles. Thus, the threshold value to distinguish particles from background noise was customized as I_thr_ = max(I_s_(t))/3. Based on the assumptions above, the real-time current signal was processed by a customized threshold as I_thr_. The current after the filter was shown in Figure 5b. It provides the current signals of standard latex particles passing through the electrodes. From Figure 5b, there were a total of 119 particles which passed through the electrodes in three seconds. Figure 5c,d provide the current response of 10 μm and 5 μm particles, respectively. From Figure 5c,d, there were 29 bigger particles and 90 smaller particles passing through the microelectrodes in three seconds. On the other hand, the expected particle counts, which were estimated by the microfluidic flows from the inlets, were about 120 and 180, respectively. The particle classification and counting experiments were repeated 5 times in order to obtain the relationship between the experimental particle counts and the estimated values. The ratio K, which expressed as N_est_/N_exp_, was calculated by K = AVE (N_i_)/n, (i = 1 ~ 5). Here N_est_ stands for the particle counts estimated by the flow rate and particle concentrations of input fluids, while N_exp_ stands for the particle counts obtained from the real-time current response obtained by the impedance analyzer. From the experimental data obtained by the impedance analyzer using the micro-casting fabricated metal-in-PDMS device, the ratio was calculated as (K_10_, K_5_) = (2.21, 4.15). 

Compared with 2D microelectrode array deposited on a silicon/glass substrate, the metal-in-PDMS microelectrodes was more like a 2.5D pattern array because the casted microelectrodes had a large height–width ratio. The thickness of a 2D microelectrode is usually at the nanoscale. However, the micro-casted microelectrodes fabricated in this work were at the micron-scale. The height of the electrodes plays important roles in electric field distributions. Salari studied the effects of electrode heights on electric field distribution and, thus, ACET (alternating current electrothermal) flows [24,25]. Our previous work also showed that electrode height would change the sensitivity of biosensors [26]. In particular, the electrode height could influence the pathway of the electric field and, thus, the amplitudes of electrical current when a particle passed through a pair of electrodes. The current signal depends on the particles size, permittivity, conductivity, as well as the position of the particles. For one single particle, the current signal was well dependent on its position. The current response of particles passing along with the central line was larger than those signals from the particles near the channel walls. In some cases, the current response may be too small to be picked up by the customized data process filter. As a result, it was always shown that the particle counts from the experimental tests were smaller than the estimated values calculated by the particle concentration of the fluids and flow rate. From the experiments above, the calibration coefficient of 10 μm particles, K_10_, was smaller than the coefficient of 5 μm particles, K_5_. It then makes sense that some of the “passing through” signals were cut off if they were smaller than the threshold values.

The proof of concept tests for commercial latex particle classification and counting applications demonstrated that the micro-casted electrode was able to provide reliable information for impedance cytometer applications. Although the coefficient K may vary with particle materials or sizes, it could be easily obtained by a group of tests. 

### 3.4. Application in Abrasive Grains Detections

As mentioned above, a group of pre-calibration tests were taken to obtain the coefficient K for abrasive grain detection. The purpose of pre-calibration tests is to reveal the relationship between the electrical signal and the particle size. Standard ferric grains were put into fresh new engine oil to prepare a standard “man-made oil” for calibration tests. Then, the particle size distribution (real data) of the “man-made oil” was tested using commercial equipment (LaserNet 220, Spectro Inc., Chelmsford, MA, US). Next, the electrical signals were recorded by the micro-casting fabricated device in this work. Also, the probability statistics of the electrical signals were obtained. By now, the relationship between the real data of particle size distribution and the electrical signal amplitude were estimated from the calibration tests. Also, the counts of abrasive grains in a cup of dirty oil from a local garage were tested using the commercial equipment and the impedance cytometer device fabricated in this work. Success in distinguishing particles can be achieved using four-terminal microelectrode arrays fabricated by the micro-casting technique presented in Section 2. As shown in Figure 6, the experimental results of the abrasive grains in the engine oil were investigated. Figure 6a provides the real-time electrical current measured by the impedance analyzer and the microelectrodes fabricated by micro-casting procedures. 

In order to investigate the sizes of these abrasive grains, two threshold values were utilized in this work. The first one, I_n1_, was defined as described in Equation (1), and it can distinguish the signals generated by particles from the background noise. With the customized thresholds, it was possible to obtain the particle counts within the microfluidic device. Figure 6c shows that there were approximately 70 particles with diameters of 15 ± 5 μm passing through the microelectrodes. Figure 6d shows that there were approximately 67 particles with diameters of 15 ± 5 μm passing through the microelectrodes. It can also be seen that the size of the abrasive particles was about 15 μm. Both the results indicate that there were detectable abrasive particles in the engine oil; thus, the engine oil should be replaced to ensure driving safety.

## 4. Conclusions

In this work, detailed experimental studies were performed to investigate the pressure-driven micro-casting fabrication of metal-in-PDMS electrodes with a large height–width ratio. The effect of surface treatment was conducted to verify the importance of surface energy control for micro-casting techniques. Fabrication failures such as flow discontinuity or metal residual occurred when the inner walls were not treated by freshly prepared reagents. In addition, the effect of inlet dimensions on the fabrication performance was also studied for d = 0.6 mm and d = 1 mm conditions. In general, a smaller inlet hole can help the melted liquid reach a smooth flow in the microchannels. As a proof of concept, particles with different sizes could be recognized by the impedance cytometer system that consisted of a commercial impedance analyzer and a microelectrode array fabricated by micro-casting technique presented in this work. As an application, it was found that the number of abrasive particles in the engine oil could be analyzed, providing useful information on abrasive particles in industrial applications.

## Figures and Tables

**Figure 1 materials-12-03710-f001:**
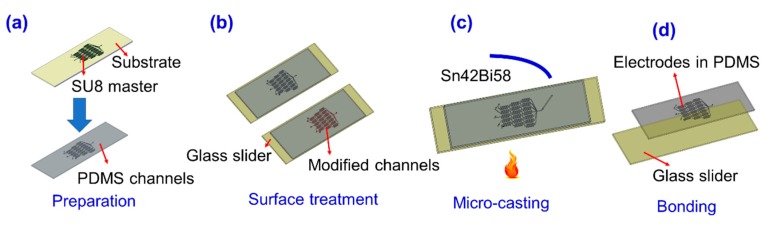
The procedures to fabricate the high height–width ratio electrodes. (**a**) Preparation of PDMS microfluidic channels. A SU8 mold was firstly lithographed, and then a PDMS layer with a patterned channel was peeled off as the mold for the micro electrodes. (**b**) surface treatment of inner walls of the microfluidic channel walls. (**c**) reversable bonding of the PDMS channel with a glass slider; melting and micro-casting within the microchannels. (**d**) In-reversable bonding of the metal-in-PDMS electrode and glass slider.

**Figure 2 materials-12-03710-f002:**
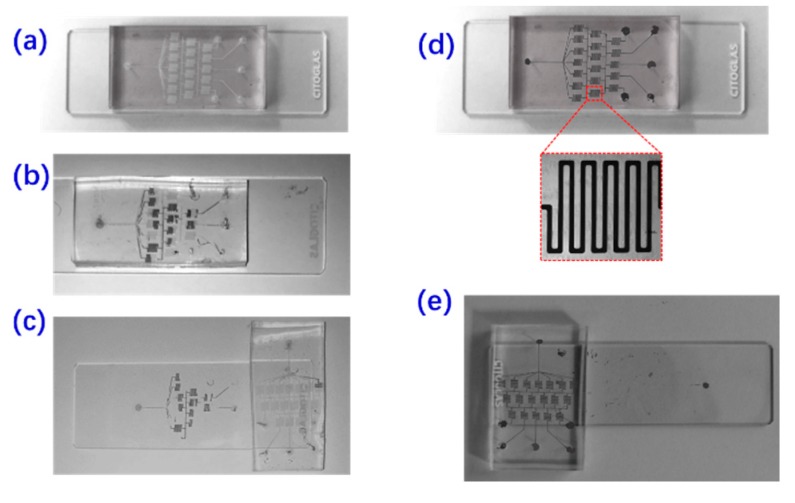
Effect of the surface treatment on the micro-casting fabrication performance. (**a**) The patterned PDMS microchannel as the casting mold; (**b**) the microchannel failed to be completely filled with the melting metal if the inner wall was not treated with surfactant; (**c**) the metal-in-PDMS electrode pattern layer failed to be peeled off from the glass substrate if the inner wall was not treated with surfactant; (**d**) the microchannel was completely filled with the melting metal, if the inner wall was well treated; (**e**) the metal-in-PDMS electrode pattern layer was peeled off from the glass substrate, if the inner wall was well treated with surfactant.

**Figure 3 materials-12-03710-f003:**
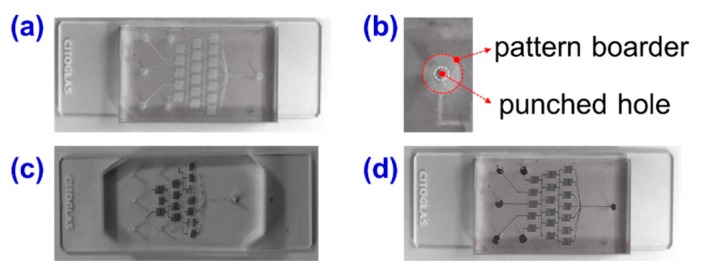
Effect of the inlet dimensions on the micro-casting fabrication performance. (**a**) The PDMS microchannel as the casting mold; (**b**) the appearance of the designed pattern border of the inlet and punched hole; (**c**) the microchannel failed to be filled with the melting metal with an inlet size of 1 mm; (**d**) the microchannel filled with melting metal with the inlet size of 0.6 mm.

**Figure 4 materials-12-03710-f004:**
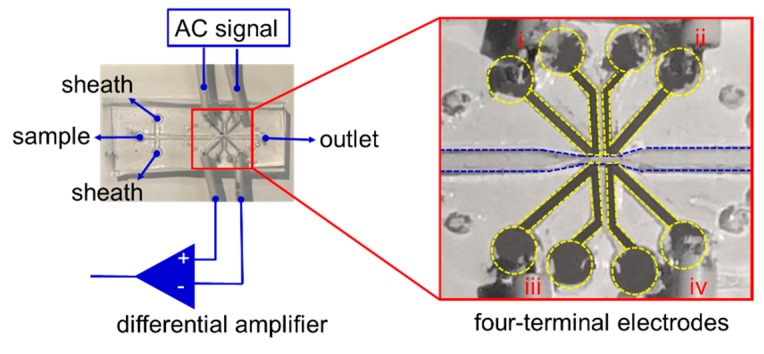
The microfluidic device used in the experiments for particle counting applications.

**Figure 5 materials-12-03710-f005:**
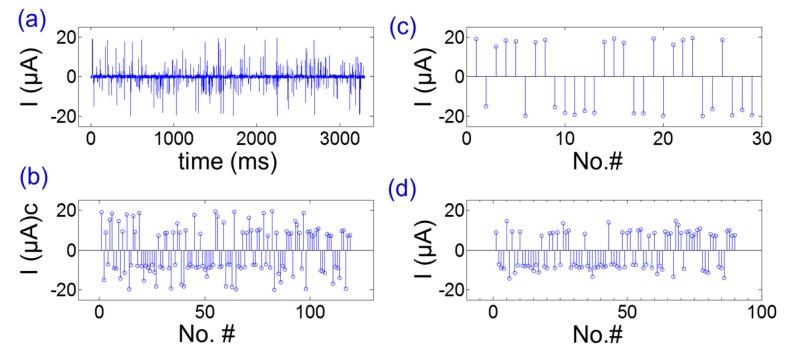
The experimental results of latex particle counting applications as a proof of concept. (**a**) The real-time current signal obtained from the impedance analyzer. (**b**) Real-time current signals processed by a customized filter. Here, the signals smaller than the threshold were cut off as zero. (**c**) The current signal response of 10 μm particles. (**d**) The current signal response of 5 μm particles.

**Figure 6 materials-12-03710-f006:**
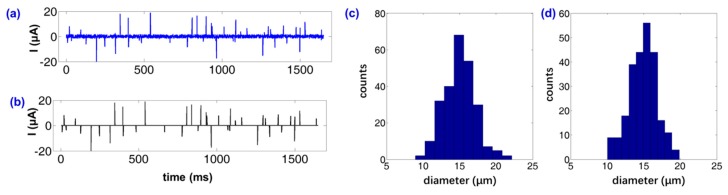
Experimental results of abrasive grains detection. Both (**a**) and (**b**) provide an example of abrasive grain detection. (**a**) The real-time current signal obtained from the impedance analyzer. (**b**) Real-time current signal processed by a customized filter. Here, the signals smaller than the I_n_ (the threshold current to distinguish the particle signals from background noise) were cut off as zero. Both (**c**) and (**d**) present the distribution of abrasive grains in the oil, obtained by the commercial equipment and the device fabricated in this work.

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
