# Peer review of "Pressure-Driven Micro-Casting for Electrode Fabrication and Its Applications in Wear Grain Detections"

_materials, 2019, doi:10.3390/ma12223710_

Round 1
Reviewer 1 Report
This paper reports a method to fabricate a microelectrode array with a high height-width ratio by using micro casting techniques and demonstration for the particle detection application. An accurate method to prepare microelectrode array is significantly important in the sensing applications. They found the effective surface treatment and the importance of the inlet dimension size for casting of the melting metal, i.e., microelectrode array fabrication in PDMS. Data in this paper will be useful for readers to prepare a microelectrode array in a high quality. Hence, I suggest that this paper is suitable for publication as Technical Note in materials after the following points are addressed.
Main points:
A high height-width ratio microelectrode array: Authors described their microelectrode has high height-width ratio. However, “how high” is not clearly mentioned. This information is essential to discuss the sensitivity of the present array in the particle detection application. If this is available, please mention in the manuscript.
Introduction: The issue to be solved in microelectrode casting should be more precisely described. This would make the significance of the paper clearer.
Surface treatment: Authors clearly showed the effect of surface treatment by the mixed reagents. From the experimental section, it seems that only one mixture of reagents were examined. Does the reagents ratio affect the properties on micro casting etc.? How the present reagent mixture ratio is important? Why the conventional plasma treatment does not work well? These should be clearly described because this section is a key in this study, and be useful information to readers.
Microfluidic device: Details of the microfluidic device consists of two PDMS layers were not easy to understand, especially the flow channel(s). Please more clearly describe this. The images in Figure 4 should also be clearer and/or higher resolution images are preferable.
Particle counting: It should be mentioned that how the current signal depends on the particle size and position of the particles and that what determine the positive or negative current. These are beneficial for readers to understand this application.
Abrasive grains detection: It should be described that how the diameter distribution was estimated in detail, so that readers understand the phenomena more easily.
Minor points
Figure 3 caption: 1mm and 0.6 mm is vise versa in the text? And missing (d).
Conclusions: In this work, detailed experimental studies were performed in this work …
Reviewer 2 Report
All in all, this is a very sound and well described work with a novel method for fabricating microelectrode structures from "liquid metal casting", using a low-temperature soldering alloy and a PDMS mould.
I have the following questions, suggestions, and recommendations:
Personally, I believe that the use of the term MEMS misused in this particular instance. There are no mechanical moving parts in the device and I find the relation to MEMS a bit too far-fetched. I would, therefore, suggest omitting the term MEMS and adapting the introduction accordingly. The authors describe a fabrication of electrode arrays on a microscale, yet, neither microscopy or SEM close-up images nor profiles using eitherprofilometry (optical or tip-based) are presented. These data needs to be provided. The electrode arrangement and the measurement principle for the particle detection experiments need to be briefly introduced to avoid misinterpretation of data.
